# Diabetic macular edema: Safe and effective treatment with intravitreal triamcinolone acetonide (Taioftal)

**Francesco Saverio Sorrentino**[1]*, **Claudio Bonifazzi**[2], **Francesco Parmeggiani**[3]

**1** Department of Surgical Sciences, Unit of Ophthalmology, Ospedale Maggiore, Bologna, Italy, **2** Department of Biomedical Sciences, University of Ferrara, Ferrara, Italy, **3** Division of Ophthalmology, Department of Biomedical and Surgical Sciences, University of Ferrara, Ferrara, Italy

\* dr.fsorrentino@gmail.com

## Abstract

### Purpose

To suggest the safety and efficacy of preservative-free triamcinolone acetonide intravitreal injectable suspension (Taioftal) for the treatment of diabetic macular edema.

### Methods

A prospective clinical study involved 49 patients (49 eyes), that were treated with Taioftal and followed-up for six months. Complete ophthalmic examination, including spectral domain optical coherence tomography, was performed at baseline, and at month 1, 3, 6 after the intravitreal injection. Accurate collection and analysis of best-corrected visual acuity (BCVA), central foveal thickness (CFT), intraocular pressure (IOP), and adverse events (AEs) were carried out in order to evaluate visual function and macular morphology before and after treatment

### Results

Median BCVA value chosen as comparing statistics was significantly improved at every follow-up time points (gain of 6 letters at month 1, 12 at month 3 –improvement up to 24% at month 3 with stabilization until month 6) compared to baseline, as certified by Kruskal-Wallis rank sum test (P<0.05). Median CFT significantly waned at each follow-up times (decrease of about 65 μm at month 1, 155 at month 3 –reduction up to 28% at month 3 keeping good outcome until month 6) compared to baseline (P<0.05). IOP elevation, with no severe increases, was the most common among spotted AEs (median of 23 mmHg at month 1, 20 at month 3).

### Conclusion

Intravitreal injection of preservative-free triamcinolone (Taioftal) is an effective, safe and inexpensive drug used to improve visual acuity and reduce central foveal thickness in eyes affected by diabetic macular edema during an average time of 6 months. Temporary, never

**Data Availability Statement:** All relevant data are within the manuscript and its files.

**Funding:** The author received no specific funding for this work.

 

**Competing interests:** The author has declared that no competing interests exist.

severe, elevation of IOP is totally manageable with topical medications. No serious vision-threatening complications are related to the use of intravitreal triamcinolone injections.

## Introduction

Macular edema (ME), continuously rising worldwide, is the main cause of visual loss in patients suffering from diabetic retinopathy (DR). Diabetic macular edema (DME), occurring at any stage of DR and often bilaterallyhttp://www.ncbi.nlm.nih.gov/pmc/articles/ PMC4458661/ - b1, results from defects in the blood-retinal barrier which lead to vascular leakage, intraretinally/subretinally fluid accumulation and macular thickening [1, 2]. The more frequent symptoms are blurring vision and drop of the best-corrected visual acuity (BCVA). The exact pathophysiology of the formation and the recurrence of ME is definitely complex [3]. Many pro-inflammatory molecules and biochemical factors are largely involved in the breakdown of the blood-retinal barrier [4].

Standard care for DME includes monitoring glycemia/glycated hemoglobin levels and focal/grid laser photocoagulation of leaking microaneurysms and areas of diffuse capillary bed leakage [3]. Clinical evidence, after recent clinical trials, has showed that anti-vascular endo-thelial growth factor therapy can be regarded as first-line therapy, but intravitreal steroids maintain a remarkable role in the management of DME [3, 5, 6]. Among steroid drugs, triam-cinolone acetonide (TA) has the effect of anti-inflammatory and anti-angiogenic agent. Many reports have demonstrated the usefulness of intravitreal TA (IVTA) in suspension form, cur-rently available as several commercial preparations, to treat DME [7–11]. TA is a synthetic ste-roid with anti-inflammatory strength five-folds higher than hydrocortisone and it has got a long-acting profile due to its low water solubility. As literature reports, the therapeutic effect of intravitreal 4 mg TA persists for up to 3 months [12, 13]. Most of studies have been carried out with TA with preservatives, but we will focus on TA preservative-free formulation, since it has been demonstrated that preservatives such as benzyl alcohol could be toxic on retinal tissues to a certain extent [14–16].

Diabetic Retinopathy Clinical Research Network recommends IVTA either alone or in combination with laser therapy in selected patients with persistent and refractory DME and vision loss.

With this study, we would like to present our experience which has highlighted a possible duration of IVTA (Taioftal) for up to 4–6 months, with a safe drug profile during the follow-up period. We have used statistical criteria to analyze changes in both visual acuity and macu-lar thickness at different stages of follow-up.

## Materials and methods

This prospective clinical study, including a total of 49 eyes of 49 patients, was conducted from May 2020 to December 2020 in the Department of Ophthalmology at Ospedale Maggiore "C. A. Pizzardi" Bologna, Italy, where the use of IVTA was regarded as therapeutic option, pre-scribed at the discretion of the treating ophthalmologist, for diabetic macular edema. This study was approved by our Institutional Review Board (Bologna Hospital Local Ethic Commis-sion) and patients signed informed consent for the use of their data. Also patients who declined to take part in this study could receive the same treatment, if the treating ophthalmol-ogist chose TA for them. The study adhered to the tenets of the Declaration of Helsinki.

The number of 49 patients with ME secondary to DR was the representative sample of a population affected by DME. Criteria for inclusion were the presence of ME secondary to DR and the central foveal thickness (CFT) higher than 300 μm as measured by spectral-domain optical coherence tomography (SD-OCT) at baseline examination. Regardless of DR grading, we included DME of non proliferative diabetic retinopathy that had never treated with laser, macular grid or panretinal photocoagulation. We ruled out DME in proliferative diabetic retinopathy. Exclusion criteria were ME secondary to other causes (retinal vein occlusion, age-related macular degeneration, uveitis, post-phaco) and eyes which already had any surgical treatment.

At baseline, all patients had a complete ophthalmic examination, comprising assessment of BCVA—using Early Treatment Diabetic Retinopathy Study (ETDRS) charts -, tonometry to measure intraocular pressure (IOP), biomicroscopy, fundus ophthalmoscopy and SD-OCT (Cirrus HD-OCT, Carl Zeiss Meditec, Inc., Dublin, CA, USA) with automated CFT measurements [17].

All patients were treated with a 4 mg/0.05 ml triamcinolone acetonide intravitreal injection (Taioftal, Sooft Italia S.p.A., Montegiorgio, IT) within two weeks from the baseline examination. IVTA was performed in the operation room. Taioftal was injected into the vitreous cavity through the pars plana using a single-use 27-gauge needle, after having performed a small corneal paracenthesis. Patients were treated with topical ophthalmic antibiotic 3 days before and 5 days after the treatment. After IVTA, they were given dorzolamide-timolol eye drops twice a day in the treated eye for 3 months. In the last 3-month period, we continued with IOP-lowering therapy only in case of IOP>19 mmHg. Moreover, we prescribed bromfenac eye drop twice a day for the first month.

Outcome measures (BCVA, CFT, IOP) were collected at month 1, month 3 and month 6. We also recorded adverse events during follow-up. There were no significant intraoperative complications as well as no significant postoperative troubles such as either endophthalmitis or severe intraocular inflammation. No cataract progression was observed during the study follow-up of six month. We compared the mean change in BCVA and CFT evaluating modifications in visual acuity and retinal morphology over six months of follow-up.

Since the values of BCVA and CFT showed single and grouped outliers at baseline, we considered the median, that is exactly the central value among all values and the quartile values, i.e. the common non-parametric statistics, for monitoring the improving of BCVA and CFT at each follow-up time. With respect to the common-used mean and standard deviation values, the advantage of using non-parametric statistics (median and percentiles) was to describe changes in BCVA and CFT even in presence of non-respondent patients. The IOP was even measured and described by means of the same statistical approach. Since the IOP elevation has been well-controlled by topic antiglaucoma medications (up to median of 23 mmHg at month 1), we avoided from providing graphs to describe IOP trends throughout this manuscript.

Data were analyzed with basic statics tools in R environment. The descriptive analysis was performed by using the box-and-whiskers plot to display outcomes (BCVA and CFT). Mean value and CI 95% were adopted to sum-up and compare the baseline with values of each follow-up time. For both parameters, the pair-wise t-test with Bonferroni correction was used to compare the baseline and the subsequent follow-up values. The value of P<0.05 was considered for statistical significance [18].

## Results

Forty-nine eyes of 49 patients (of age 45–90, median 75 years) with DME undergoing a triamcinolone injection have been included for this analysis.

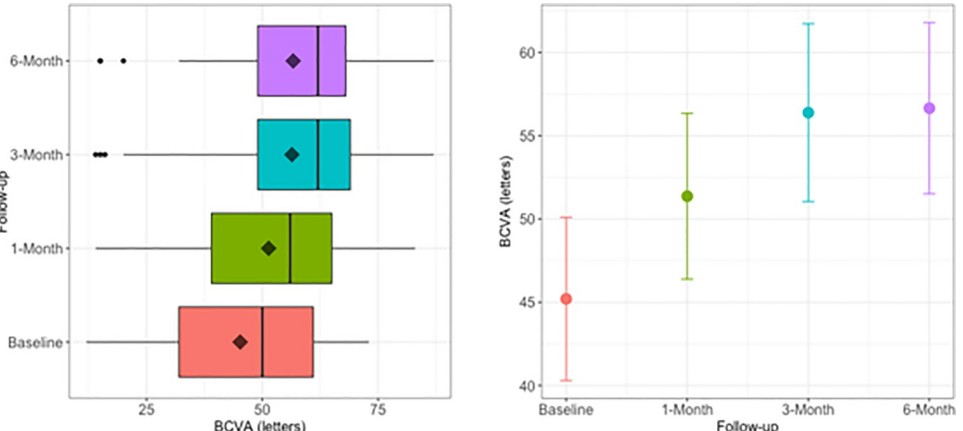

**Fig 1. A.** Box-plot of best-corrected visual acuity (BCVA) at baseline and each time of the study follow-up (month 1, 3 and 6). The BCVA median, depicted as vertical bar in each box, strongly increases till the month 3. It substantially remains unchanged at month 6. The BCVA mean, depicted as little rhombus in each box, progressively increases till the month 3, remaining more or less unvaried at month 6. Due to the presence of outliers, the mean values are systematically lower than the median values. **B.** Mean±95% CI of best-corrected visual acuity (BCVA) at baseline and each time of the study follow-up (month 1, 3 and 6). The plot shows the increasing trend of mean BCVA with the related 95% CI. The increasing trend underlines the overall rise of BCVA during the period of maximum effectiveness of IVTA (month 3). The graph shows the steady level of BCVA at month 6. As evident, at month 3 and 6 the 95% CI does not overlap baseline. This highlights the fact that we have a remarkable and lasting effect over the time. The BCVA percentile values and the trend over the follow-up period are detailed in Table 1.

## Visual acuity

At baseline, the median of BCVA was 50 letters (12 to 73 as range). It progressively improved up to 56 letters (14 to 83) at month 1, 62 (14 to 87) at month 3, and remained 62 (15 to 87) at month 6. The Bonferroni pairwise test gave a statistical significance (P<0.01) for each comparison follow-up time vs baseline. As expected, it was not significant between month 6 and month 3 (P = 0.9) of follow-up.

The dispersion of values of BCVA (Fig 1A and 1B, Table 1) is represented as difference between Q3 e Q1 percentile values (i.e., Interquartile range, IQR). At month 1, the dispersion of central values remains similar to baseline. At month 3 and 6, the dispersion of central values decreases (lower IQR with respect to baseline and month 1). On Fig 1A, specifically at month 3 and 6, there are some outliers that precisely indicate patients, for whom treatment was not effective.

The Table 1 displays a similar minimum value at both baseline and follow-up period. On the contrary, the percentiles from Q1 to maximum value have an increasing of more than 10 letters. Furthermore, the difference between the third (Q3) and first quartile (Q1) remarkably

**Table 1. Best-corrected visual acuity measured over the time (baseline and follow-up).**

| Time | Min | Q1 | Median | Mean | Q3 | Max | Mean | SE | 95% CI |
|---|---|---|---|---|---|---|---|---|---|
| Baseline | 12 | 32 | 50 | 45 | 61 | 73 | 45.2 | 2.4 | 4.9 |
| 1-Month | 14 | 39 | 56 | 51 | 65 | 83 | 51.4 | 2.5 | 5.0 |
| 3-Month | 14 | 49 | 62 | 56 | 69 | 87 | 56.4 | 2.7 | 5.3 |
| 6-Month | 15 | 49 | 62 | 57 | 68 | 87 | 56.7 | 2.6 | 5.1 |

Min: minimum value, Q1: percentile 25%, Q3: percentile 75%, Max: maximum value, SE: standard error, CI: confidence interval.

We inserted two-fold the mean to recall the label of Fig 1. In the left columns the percentile values are displayed, in the right columns the mean values and 95% CI.

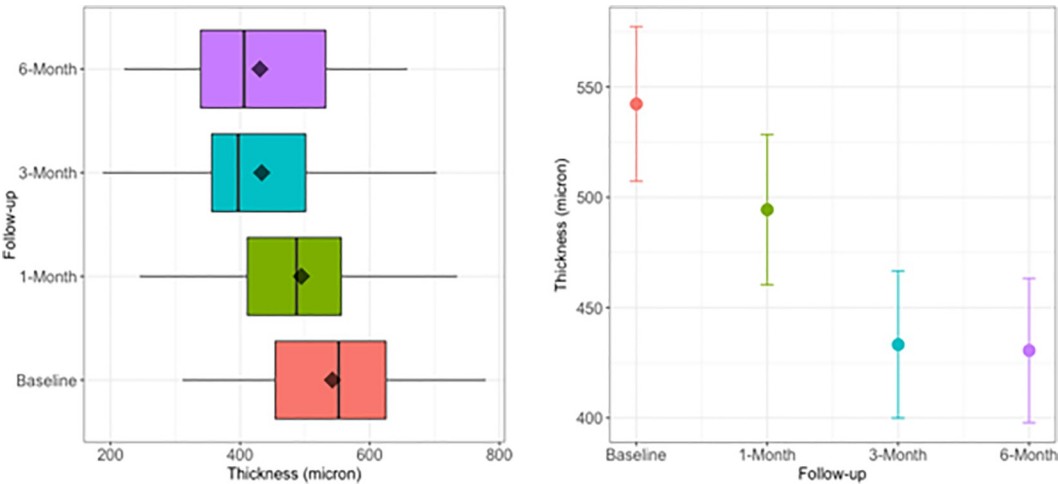

**Fig 2. A.** Box-plot of central foveal thickness (CFT) at baseline and each time of the study follow-up (month 1, 3 and 6). The CFT median, depicted as vertical bar in each box, remarkably decreases till the month 3, with a slight increasing at month 6. The CFT mean, depicted as little rhombus in each box, progressively decreases till the month 3, remaining more or less unvaried at month 6. To note, at month 3 and 6 the mean values are higher than the median, and the CFT values less than 400 µm are more grouped. **B.** Mean±95% CI of central foveal thickness (CFT) at baseline and each time of the study follow-up (month 1, 3 and 6). The plot displays the decreasing trend of CFT mean with the related 95% CI. This trend underlines the overall decrease of CFT during the period of maximum effectiveness of IVTA (month 3). The graph displays a sort of steady level of CFT at month 6. As evident, at month 3 and 6 the 95% CI does not overlap baseline. This highlights the fact that we have a remarkable and lasting effect over the time. The CFT values of 95% CI and the trend over the follow-up period are detailed in Table 2.

decreases (about 30%), see Fig 1A. To note, the mean standard error SE≈3 gives indication that there is a general improvement of BCVA for each time of follow-up.

After IVTA, the BCVA median significantly improved at month 1 and month 3, increasing respectively of 12% (6 letters at ETDRS) and 24% (12 letters at ETDRS) compared to baseline. Neither further improvements nor worsening happenings occurred at month 6, substantially remaining unvaried with respect to month 3. The non-parametric Kruskal-Wallis rank sum test certificated that there was a difference in median values at month 1, 3 and 6 with respect to baseline (P<0.01).

## Retinal morphology

At baseline, the median of CFT was 552 µm (311 to 779 as range). It progressively decreased to 487 µm (245 to 735) at month 1, 397 µm (188 to 703) at month 3 and 406 µm (221 to 658) at month 6. The Bonferroni pairwise test gave a statistical significance (P<0.01) for each comparison follow-up time vs baseline. As expected, it was not significant between month 6 and month 3 (P≈1.0) of follow-up.

After IVTA, the CFT median significantly decreased at month 1 and month 3, reducing respectively of 12% (65 µm at OCT) and 28% (155 µm) compared to baseline. At month 6, no significant changes occurred and retinal thickness remained approximately similar to month 3. The non-parametric Kruskal-Wallis rank sum test certificated that there was a difference in median values at month 1, 3 and 6 with respect to baseline (P<0.01).

As above-mentioned, the dispersion of CFT values (Fig 2A and 2B, Table 2) is represented as difference between Q3 e Q1 values (i.e., Interquartile range, IQR). At month 1 and 3 the IQR value is slightly reduced (at most 25 µm); at month 6, the dispersion of central values slightly increases. Small differences are detectable in Range value (80 µm at month 3, 30 at

**Table 2. Central foveal thickness measured over the time (baseline and follow-up).**

| Time | Min | Q1 | Median | Mean | Q3 | Max | Mean | SE | 95%CI |
|------|-----|-----|--------|------|-----|-----|------|-----|-------|
| Baseline | 311 | 454 | 552 | 542 | 625 | 779 | 542.3 | 17.9 | 35.1 |
| 1-Month | 245 | 411 | 487 | 494 | 556 | 735 | 494.4 | 17.4 | 34.1 |
| 3-Month | 188 | 356 | 397 | 433 | 501 | 703 | 433.2 | 17.0 | 33.4 |
| 6-Month | 221 | 339 | 406 | 431 | 532 | 658 | 430.6 | 16.7 | 32.7 |

Min: minimum value, Q1: percentile 25%, Q3: percentile 75%, Max: maximum value, SE: standard error, CI: confidence interval.

We inserted two-fold the mean to recall the label of Fig 2. In the left columns the percentile values are displayed, in the right columns the mean values and 95% CI.

month 6). By observing the IQR and Range values, the reduction of foveal thickness does not display such a clear trend as visual acuity does.

The Table 2 displays different minimum values for each time of follow-up with a maximum CFT decreasing of about 30%. A reduction of CFT from 15 to 25% is common to each percentile (see the quartile vertical lines in Fig 2A). As above-mentioned, the common mean standard error SE≈18 μm confirms the overall decreasing of CFT during the follow-up period (see the 95% CI width in Fig 2B).

## Correlation visual acuity vs retinal morphology

We evaluated the correlation between BCVA and CFT values. The graph in Fig 3 displays the linear regression analysis that has been built up by using values of Intercept and Slope according to the following formula: BCVA = Intercept + Slope * CFT.

For each time of follow-up, P<0.01 shows that this linear model correctly describes the existing relationship between CFT and BCVA. The coefficient of determination R-squared (R-sq) significantly enhances at month 1 and month 3 with respect to baseline, but also at month

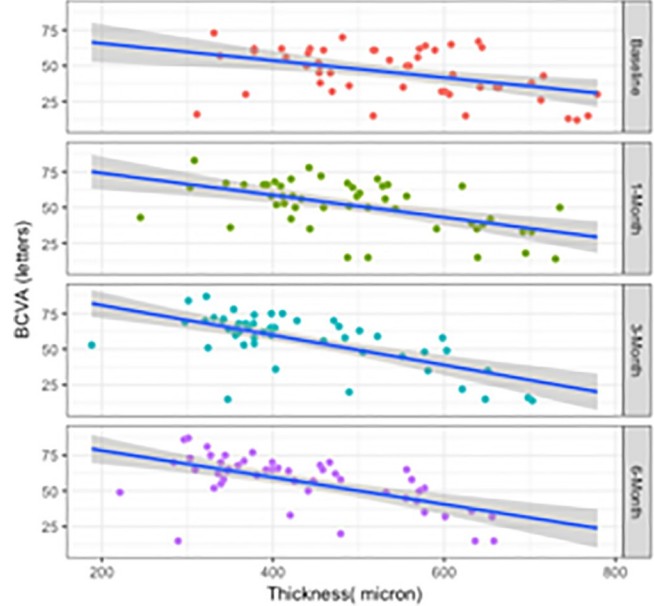

| Follow-up | Intercept | Slope |
|-----------|-----------|-------|
| Baseline | 77.74 | -0.06 |
| 1-Month | 86.67 | -0.08 |
| 3-Month | 101.74 | -0.11 |
| 6-Month | 97.10 | -0.09 |

| Follow-up | R-sq | P-value |
|-----------|------|---------|
| Baseline | 0.18 | <0.01 |
| 1-Month | 0.28 | <0.01 |
| 3-Month | 0.43 | <0.01 |
| 6-Month | 0.36 | <0.01 |

**Fig 3. Linear regression between best-corrected visual acuity (BCVA) and central foveal thickness (CFT) at baseline and each time of follow-up (month 1, 3 and 6).** The Slope absolute value increases from baseline to 6-month highlighting the linear relationship between CFT and BCVA. The improving of BCVA for decreasing CFT is clear for 3-month and 6-month, as stressed by the grouping of points (CFT-BCVA) on the left side in the bottom panels. The linear relationships at 3-month and 6-month are substantially very similar, as made evident by Intercept and Slope values.

6 it is higher than baseline and month 1. The increased R-sq indicates that values of BCVA and CFT are closer to the linear line.

Due to Intercept and Slope values, we can give a quantitative evaluation of the efficacy of IVTA over the time of 6-month follow-up. Slope is characterized by negative values for each time of follow-up to highlight the fact that decreased retinal thickness can improve the visual function. Slope indicates the increasing amount of letters of BCVA for 100 μm of CFT reduction. At baseline, Slope = -0.06 means that without IVTA we can have a gain of 6 letters for 100 μm decrease of CFT. At month 3, Slope = -0.11 means that after IVTA we have a gain of 11 letters instead of 6 for 100 μm CFT reduction. At month 6, Slope = -0.09 means gain of 9 letters for each 100 μm of persistent CFT reduction.

## Intraocular pressure

We measured IOP at baseline and at each time (month 1, 3 and 6) of follow-up. We observed a peak of IOP at month 1, with constantly decreasing trend at month 3 and month 6. The median at baseline was 16 mmHg (12 to 20 as range). At month 1 the IOP median was 23 (13 to 30), 20 (13 to 30) at month 3, and 17 (12 to 20) at month 6. The Bonferroni p-value adjustment method was statistically significant ($P<0.01$) by comparing month 1 and 3 with baseline mean values, but not significant between month 6 and baseline ($P = 0.12$).

## Discussion

In this prospective study, we evaluated progressive changes in visual acuity and macular morphology in eyes undergoing IVTA injections (Taioftal) for DME over a 6-month follow-up period. We found that the treatment with injectable preservative-free suspension of TA initially induced a significant improvement in BCVA, already one month after administration (6 letters at ETDRS, 12% of improvement). The best result was at month 3 of follow-up (12 letters at ETDRS, 24% of improvement) and the positive effect lasted approximately up to month 6. Likewise, CFT showed an ameliorating trend, significantly decreasing after three months (28% of reduction, corresponding to more than 150 μm with respect to baseline), and keeping the good outcomes until the sixth month after treatment. At month 6, CFT did not significantly get better with respect to month 3, but it displayed a sort of stabilization keeping rather good outcomes compared to baseline.

Over recent years, several studies have showed the safety and efficacy of IVTA injections in patients with DME. Our results are substantially in according with these studies, but we did not observe a drop in outcomes at month 6 [19–22]. Also, we have been positive impressed by our data at month 6: no further improved outcomes compared to month 3, but not significant worsened at all. The linear regression analysis, describing the relationship between CFT and BCVA, has showed a progressive increase of coefficient of determination R-sq. This value significantly enhances at month 1 and 3 with respect to baseline, pointing out that the improvement of BCVA fully depends on CFT during the period of utmost efficacy of Taioftal. However, the R-sq value at month 6 higher than one at baseline and month 1 confirms that the effect of IVTA is rather durable, lasting more than 3 months as medical literature usually reports.

The stable clinical condition we reached is likely due to different and concurrent conditions. As known, there is the multifaceted pharmacological effect of steroids: anti-inflammatory, anti-VEGF and immunosuppressor. Taioftal has got precise physical and chemical characteristics that make this injectable drug similar but not exactly the same of other TA formulations for intravitreal use. Distinct preservative-free TA formulations substantially differ each other in pH value, particle size and load, solubility, dissolution, pharmacodynamics, and pharmacokinetics after intravitreal injection [12, 23, 24].

So far, many studies have been carried out with the use of triamcinolone acetonide with preservatives (Kenalog®, Kenacort®), whereas only few studies have discussed the use of preservative-free triamcinolone acetonide (Triescence®). The difference between triamcinolone with or without preservatives is relevant, because the use of molecules such as benzyl alcohol could provoke toxicity on retinal tissues and also might bring to a rise of IOP [14–16]. With Taioftal the severity of ocular hypertension is negligible. There is a peak of IOP at month 1, then decreasing with no ocular complications and without affecting visual outcomes, and it is absolutely manageable with topical hypotensive medications. As above-mentioned in Methods, we performed a little paracenthesis right before the intravitreal injection and we prescribed topic therapy with IOP-lowering for three months and anti-inflammatory eye drops for the first month after IVTA. In this way, we have never had untreatable hypertone and we had well-managed peaks of IOP in the course of follow-up period. We were wondering the reason of the stability of outcomes during follow-up. Undoubtedly, TA has got a profile of safety and effectiveness for DME treatment. But, we would like to highlight how helpful bromfenac might be as supportive topic therapy in the first thirty days. BCVA maintained slightly better outcomes than CFT, which improved, gradually but less quickly, up to month 3, tending to early worsen at month 6. We observed less scattered values of BCVA and CFT at month 3 and 6 compared to baseline, therefore we can state that TA can induce a global improving effect. Nonetheless, probably there was something else that kept good visual function at month 6, while the retinal morphology slowly began to get worse again. An intriguing study carried out by Imai and coworkers reported that there is a close link between inflammation, microvascular permeability and glucocorticoid receptor in DME [25]. Thus, we assumed that IVTA injection could reduce neurodegeneration and neuroinflammation by lowering the level of inflammation in the microenvironment of the vitreous cavity as well as on the surface of the vitreoretina interface [26, 27].

In conclusion, we have showed that intravitreal preservative-free triamcinolone injections provide significant functional and morphological benefits in patients with DME. After three months, BCVA significantly improves of about 24% with respect to baseline. But this good outcome remains at month 6 with a more uniformed distribution around the median value of BCVA. Conversely, CFT significantly decreases of about 28% up to month 3 and displays a very little worsening at month 6 with much more dispersion of values, regardless of the good stability of BCVA. No troubles with the increase of IOP in the first three months, because it is well-controlled by topic IOP-lowering treatments. Further studies and investigations are needed to better evaluate the long-term effect of IVTA on DME.

This study has demonstrated that one intravitreal preservative-free triamcinolone injection accompanied by topical treatment (IOP-lowering and anti-inflammatory eye drops) and, mainly, a good balance of systemic glico-metabolic situation can be effective, well-tolerated and rather inexpensive as mid-term treatment (four-six months) against ME caused by diabetes.

## Acknowledgments

We acknowledge physicians and nurses for working at the retinal disease office and doctors for performing intravitreal injections.

## Author Contributions

**Conceptualization:** Francesco Saverio Sorrentino.

**Data curation:** Francesco Saverio Sorrentino, Claudio Bonifazzi.

**Methodology:** Francesco Saverio Sorrentino.

**Supervision:** Francesco Parmeggiani.

**Writing – original draft:** Francesco Saverio Sorrentino.

**Writing – review & editing:** Francesco Saverio Sorrentino.

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
