## [Decision Letter · Decision Letter 0]

21 Jun 2021

PONE-D-21-17143

Diabetic macular edema: safe and effective treatment with intravitreal triamcinolone acetonide (Taioftal)

PLOS ONE

Dear Dr. Sorrentino,

Thank you for submitting your manuscript to PLOS ONE. After careful consideration, we feel that it has merit but does not fully meet PLOS ONE’s publication criteria as it currently stands. Therefore, we invite you to submit a revised version of the manuscript that addresses the points raised during the review process.

The reviewers report that your findings are sound and that the results support your conclusions, but one reviewer indicates that the abstract needs more detail and both recommend additional papers for you to cite and discuss. Please distinguish between triamcinolone acetonide with preservatives preservative-free triamcinolone acetonide.

We look forward to receiving your revised manuscript.

Kind regards,

Alfred S Lewin, Ph.D.

Academic Editor

PLOS ONE

Journal Requirements:

2. Thank you for including your ethics statement:  "This study was approved by our Institutional Review Board".   

Reviewers' comments:

Reviewer's Responses to Questions

**Comments to the Author**

1. Is the manuscript technically sound, and do the data support the conclusions?

Reviewer #1: Yes

Reviewer #2: Yes

2. Has the statistical analysis been performed appropriately and rigorously? 

Reviewer #1: Yes

Reviewer #2: Yes

3. Have the authors made all data underlying the findings in their manuscript fully available?

Reviewer #1: Yes

Reviewer #2: Yes

4. Is the manuscript presented in an intelligible fashion and written in standard English?

Reviewer #1: Yes

Reviewer #2: Yes

5. Review Comments to the Author

Reviewer #1: This is a very interesting manuscript on a topic of great relevance in a real world scenario. The use of intravitreal triamcinolone acetonide has been used as primary treatment and as an adjuvant treatment to other treatment modalities in diabetic macular edema. However, the use of this steroid is not exempt from adverse events such as ocular hypertension and the appearance and/or progression of cataract.

It is important to mention that most of the studies with this drug have been with the use of triamcinolone acetonide with preservatives (Kenalog®, kenacort®) and there are few studies on the use of preservative-free triamcinolone acetonide. This is relevant, since the use of preservatives such as benzyl alcohol have shown toxicity in ocular tissues. Therefore, it is relevant for the authors to discuss this point and its possible relationship with the adverse events observed, specifically, with the rate and severity of ocular hypertension observed, and with the visual results at 3 and 6 months after the injection of the drug. There are some published articles that talk about the subject:

1. Altamirano-Vallejo J.C., Mora-Ríos L.E., Castellanos-González M.A., et al. Comparison of intraocular pressure in eyes with macular edema treated with intravitreal triamcinolone acetonide with and without preservatives. Rev Mex Oftalmol. 2008;82(6):397-402

2. Li Q, Wuang J, Yang L. et al. A Morphologic study of retinal toxicity induced by triamcinilone acetonide vehicles in rabbit eyes. Retina 2008; (28):504-10

3. Lüke M, Januschowski K, Beutel J, Warga M, Grisanti S., et al. The effects of triamcinolone crystals on retinal function in a model of isolated perfused vertebrate retina. Exp Eye Res 2008; (1):22-9.

In general, it is an interesting article that opens the possibility of making new comparative studies in the future specially between triamcinolone acetonide with and without preservatives.

Reviewer #2: Abstract could be wrote more detailed

What do ou think about the rebound phenomenon for triamcinolone? You can read this article (Dikmetas O, Kuehlewein L, Gelisken F. Rebound Phenomenon after Intravitreal Injection of Triamcinolone Acetonide for Macular Edema. Ophthalmologica. 2020;243(6):420-425. doi: 10.1159/000507712. Epub 2020 Apr 6. PMID: 32252055.)

Hoe do you make the analysis of the data? Only one researcher? intraobserver reliability?

Table and figures are too busy. I think you can make them more understandable

6. PLOS authors have the option to publish the peer review history of their article (what does this mean?). If published, this will include your full peer review and any attached files.

Reviewer #1: **Yes: **Arturo Santos MD Ph.D.

Reviewer #2: No

---

## [Author Response · Author response to Decision Letter 0]

6 Aug 2021

First of all, I thank you all for the time dedicating to evaluate this manuscript giving your personal scientifically considerations. It has been a pleasure to revise it, according to your careful observations and useful suggestions. 

To Reviewer 1, you were perfectly right to underline the fact I would have stressed about the difference between triamcinolone with preservatives and preservative-free formulation. I focused on a specific preservative-free molecule (Taioftal). According with papers you pointed out, I enriched this manuscript with considerations about the difference between treatments with preservative or preservative-free triamcinolone. I totally agree about the next chance to make comparative studies between different types of triamcinolone.

To Reviewer 2, I rewrote the abstract adding some important data as you requested. I collected all data during my routinely outpatient clinics, so there was just me as researcher. However, the statistical analysis was performed with the essential help of Professor Claudio Bonifazzi from the University of Ferrara. Also, the design of the study is due to Professor Francesco Parmeggiani from the University of Ferrara. I really apologize for my shameful absent-mindedness about the list of Authors. About Figures and Tables, I think there are not busy but rich of information and data. With a deeper look at graphs and legends, you can understand the sense of the entire paper. Anyway, I tried to explain figures as much as I can because I recognize their complexity. Thank you for the interesting paper you reported about the rebound phenomenon after IVTA. Authors pointed out this phenomenon from baseline at 2 months after IVTA injection. Personally, I have never experienced it when treating diabetic macular edema, but rarely when treating macular edema post-occlusion.

Thank you all again for your help to improve this manuscript.

---

## [Editor Report · Decision Letter 1]

8 Sep 2021

Diabetic macular edema: safe and effective treatment with intravitreal triamcinolone acetonide (Taioftal)

PONE-D-21-17143R1

Dear Dr. Sorrentino,

We’re pleased to inform you that your manuscript has been judged scientifically suitable for publication and will be formally accepted for publication once it meets all outstanding technical requirements.

Kind regards,

Alfred S Lewin, Ph.D.

Section Editor

PLOS ONE
---

## [Editor Report · Acceptance letter]

23 Sep 2021

PONE-D-21-17143R1 

Diabetic macular edema: safe and effective treatment with intravitreal triamcinolone acetonide (Taioftal) 

Dear Dr. Sorrentino:

I'm pleased to inform you that your manuscript has been deemed suitable for publication in PLOS ONE. Congratulations! Your manuscript is now with our production department. 

Kind regards, 

on behalf of

Dr. Alfred S Lewin 

Section Editor

PLOS ONE